# The pathogenesis of experimental Emergomycosis in mice

**Maxine A. Höft**[1,2], **Lucian Duvenage**[1,2], **Sumayah Salie**[1,2], **Roanne Keeton**[2], **Alfred Botha**[3], **Ilan S. Schwartz**[4], **Nelesh P. Govender**[1,5,6], **Gordon D. Brown**[1,6], **Jennifer Claire Hoving**[1,2,6]*

**1** CMM AFRICA Medical Mycology Research Unit, Institute of Infectious Diseases and Molecular Medicine (IDM), University of Cape Town, Cape Town, South Africa, **2** Division of Immunology, Department of Pathology, Faculty of Health Sciences, University of Cape Town, Cape Town, South Africa, **3** Department of Microbiology, Stellenbosch University, Stellenbosch, South Africa, **4** Division of Infectious Diseases, Department of Medicine, Duke University School of Medicine, Durham, North Carolina, United States of America, **5** National Institute for Communicable Diseases, Division of the National Health Laboratory Service, Johannesburg, South Africa; School of Pathology, Faculty of Health Sciences, University of the Witwatersrand, Johannesburg, South Africa, **6** Medical Research Council Centre for Medical Mycology at the University of Exeter, Geoffrey Pope Building Stocker Road, Exeter, United Kingdom

* jennifer.hoving@uct.ac.za

**Data Availability Statement:** All relevant data are within the manuscript and its Supporting Information files.

**Funding:** This work was supported by the National Research Foundation of South Africa (to JCH)

## Abstract

*Emergomyces africanus* is a recently identified thermally-dimorphic fungal pathogen that causes disseminated infection in people living with advanced HIV disease. Known as emergomycosis, this disseminated disease is associated with very high case fatality rates. Over the last decade, improved diagnostics and fungal identification in South Africa resulted in a dramatic increase in the number of reported cases. Although the true burden of disease is still unknown, emergomycosis is among the most frequently diagnosed dimorphic fungal infections in Southern Africa; and additional species in the genus have been identified on four continents. Little is known about the pathogenesis and the host's immune response to this emerging pathogen. Therefore, we established a murine model of pulmonary infection using a clinical isolate, *E. africanus* (CBS 136260). Both conidia and yeast forms caused pulmonary and disseminated infection in mice with organisms isolated in culture from lung, spleen, liver, and kidney. Wild-type C57BL/6 mice demonstrated a drop in body weight at two weeks post-infection, corresponding to a peak in fungal burden in the lung, spleen, liver, and kidney. An increase in pro-inflammatory cytokine production was detected in homogenized lung supernatants including IFN-γ, IL-1β, IL-6, IL12-p40 and IL-17 at three- and four-weeks post-infection. No significant differences in TNF, IL-12p70 and IL-10 were observed in wild-type mice between one and four-weeks post-infection. Rag-1-deficient mice, lacking mature T-and B-cells, had an increased fungal burden associated with reduced IFN-γ production. Together our data support a protective T-helper type-1 immune response to *E. africanus* infection. This may provide a possible explanation for the susceptibility of only a subset of people living with advanced HIV disease despite hypothesized widespread environmental exposure. In summary, we have established a novel murine model of *E. africanus* disease providing critical insights into the host immune components required for eliminating the infection.

Carnegie Corporation (to JCH), Wellcome Trust (209293 to GDB and 217163 to JCH), the Medical Research Council (UK) Centre for Medical Mycology (MR/N006364/2) (to GDB), the Oppenheimer Memorial Trust (to MAH) and the University Research Council, University of Cape Town (to LD). The funders played no role in the study design, data collection and analysis, decision to publish, or preparation of the manuscript.

**Competing interests:** The authors have declared that no competing interests exist.

## Author summary

*Emergomyces africanus* is a thermally-dimorphic fungus that was recently described as the cause of disseminated infections in persons living with advanced HIV disease in South Africa, where emergomycosis is among the most frequently diagnosed dimorphic fungal diseases. Four additional *Emergomyces* species have been described causing serious infections around the world, and the infection has a high case fatality rate. Despite the seriousness of this infection, the pathophysiology, immunology, diagnosis, and optimal management of emergomycosis remain largely unknown, hampered by the absence of an animal model. Here, we established a mouse model of pulmonary and disseminated emergomycosis using a clinical isolate of *E. africanus* with the aim of investigating the host immune response. We found that infection spread from the lungs to other organs within two weeks. We describe the effector immune responses in immune-competent mice aiding in controlled infection. We also highlight the immune-modulating components lacking in immunocompromised mice that promote susceptibilty to infection. Our study provides new insights into the components of the host immune response required for controlling *Emergomyces* spp. infection.

## Introduction

*Emergomyces africanus* is a thermally-dimorphic fungal pathogen belonging to the Ajellomycetaceae (Onygenales) family [1]. Emergomycosis is the disease caused by seven dimorphic fungal species belonging to the newly described genus, *Emergomyces* [2,3]. These include *Emergomyces pasteurianus* (Europe, Asia, Africa), *E. canadensis* (North America), *E. orientalis* (Asia) and *E. europaeus* (Europe) [2]. Recently, *E. sola* and *E. crescens* have been included into the *Emergomyces* genus [3]. Molecular evidence suggests that *E. africanus* is found in the environment. Using spore traps (for air) and a PCR assay, DNA was detected in air and soil samples [4,5]. Hyphae produce conidiophores and conidia which are thought to be the infectious agent whereby conidia and hyphal fragments are aerosolised upon soil disruption or wind and inhaled by the host [5]. Once inhaled, the change in temperature from environmental to body temperature (37˚C) is believed to drive a morphological switch from conidia to a yeast-like phase, which is responsible for pulmonary and upon dissemination, extrapulmonary disease [6]. 2013).

To date, almost all reported cases of emergomycosis are opportunistic infections, occurring among immunocompromised individuals with the majority of cases in persons with advanced HIV disease [6,7]. Radiographic findings of pulmonary disease were present in 86% of South African patients at the time of diagnosis [8], although most patients are diagnosed late in disease course, after the fungus has disseminated to skin. Cutaneous lesions, appearing as papules, plaques, nodules or ulcers, with widespread distribution, have been described in >96% of patients at the time of diagnosis [8]. Dissemination to other tissues such as the gastrointestinal tract, liver, lymph nodes and bone marrow also occurs but are more difficult to diagnose [8].

Diagnostic methods include fungal culture from tissue biopsy as well as histopathology. Positive histopathology with special fungal stains results in the visualisation of small yeast-like cells (2–4 μm diameter) with narrow-based budding [6]. *Emergomyces* spp. can often be misidentified as *Histoplasma capsulatum* as the histopathology findings are very similar [9]. Polymerase chain reaction (PCR) amplification of fungal DNA in infected tissue can also be used as a diagnostic method by amplifying regions of the nuclear ribosomal internal transcribed

spacer (ITS) and β-tubulin gene [2,10]. There are currently no sensitive and specific serological or antigen tests for emergomycosis and there is an urgent need for a rapid and cost-effective diagnostic that may be available in resource-limited settings.

Emergomycosis is assumed to be universally fatal if not treated, while case-fatality ratios in patients receiving treatment are still high, possibly due to diagnosis occurring at late stages of infection. There are consensus guidelines for the diagnosis and treatment of emergomycosis but many of the recommendations are extrapolated from histoplasmosis treatment approaches due to limited human and absent animal data [11]. There are currently no animal models of pulmonary infection with emergomycosis. In a proof-of-principle study, Schwartz et al infected mice via intraperitoneal inoculation. However, this would not represent natural infection believed to be via inhalation and therefore limits the understanding of disease kinetics and immunologic response during infection [4]. The development of an animal model of pulmonary emergomycosis to study the pathophysiology and immunology of infection and advance diagnostics and treatment strategies is therefore an urgent priority.

The goal of this study was to establish a mouse model of pulmonary disease and to identify key immune components during infection. Using a clinical isolate of *E. africanus*, we were able to induce a pulmonary and disseminated infection in wild-type mice. We showed that following infection, *E. africanus*, disseminated from the lung to the spleen, liver, and kidneys within two weeks; and by four weeks, it had been cleared from the lung, liver, and kidney (but not spleen). Infection triggered a T helper-type 1 immune response supported by the detection of pro-inflammatory cytokines such as IFN-γ, IL-1β, IL-6, IL-12p40 and IL-17 in the supernatant of lung homogenates. These observations were further elucidated by conducting studies involving immunocompromised mice (Rag-1$^{-/-}$), which lack mature T and B cells. These studies revealed an increased fungal burden coupled with diminished IFN-γ production.

Understanding the kinetics of *E. africanus* in an immunocompetent murine model provides insight into the vital immune components orchestrating control and clearance of infection. This model creates the foundation for future mechanistic studies of disease progression and clearance.

## Methods

### Ethics statement

All the experiments performed were conducted in accordance with the Animal Research Ethics Committee of South African National Standard (SANS 10386:2008) and UCT, South Africa for the practice of animal procedures. The protocols were approved by the Animal Ethics Committee (permitted ethics protocol numbers AEC: 017/001 and 020/017), Faculty of Health Sciences, UCT, South Africa. Individuals involved in the study were accredited by the South African Veterinary Council for all animal experimental procedures and welfare monitoring.

### Mouse strains

Male and female mice between the age of 8–12 weeks (age and sex-matched), were used during *in vivo* experiments. Wild-type mice of different genetic backgrounds: C57BL/6, BALB/c and Sv129 mice from Jackson Laboratories were used. Disease progression and immune responses in wild-type C57BL/6 mice were compared with the immunocompromised Rag-1$^{-/-}$ mouse strain on a C57BL/6 background (Jackson Laboratories). All mice were housed in specific pathogen-free conditions in individually ventilated cages within the University of Cape Town biosafety level 2 (BSL2) animal unit facility.

### *Emergomyces africanus* strain and growth conditions

A clinical isolate of *E. africanus* (CBS 136260; accession no JX398291) was acquired from Nelesh Govender (National Institute for Communicable Diseases, South Africa) and was used in all experiments [6]. Laboratory culture of *E. africanus* was performed as described by Kenyon et al., 2013 within biosafety level 2 facilities. Briefly, *E. africanus* was grown in the filamentous form at 25–29˚C on Difco TM Sabouraud Dextrose Agar (SDA) (Difco Laboratories; Becton Dickinson, and Company (BD)) and in the yeast form at 37˚C on brain heart infusion (BHI) agar (Sigma-Aldrich, USA). The yeast phase suspension was propagated by inoculating BHI broth followed by incubation at 37˚C with agitation at 180 RPM and harvested at a mid-log phase. Sterile aliquots of inoculum were stored at -80˚C in BHI broth with 30% glycerol, after counting the cells for further use. The stock concentration of fungal cells was determined by serial dilution and plating of randomly selected vials.

### *Emergomyces africanus* murine infection model

*E. africanus* yeast inoculum was thawed and washed by centrifugation at 6000 RPM for 10 minutes (min) and the pellet was resuspended in sterile PBS. A serial dilution was performed to obtain the desired concentration of cells per 50 μl directly before infection. Mice were anaesthetized by intraperitoneal injection with ketamine and xylazine (80 mg/kg and 16 mg/kg respectively) and 50 μl of the inoculum was administered via non-invasive intratracheal inoculation. Inspection for humane endpoints in sick mice was highly regarded and adhered to during the study. Mice showing weight loss greater than 20% of their baseline weight, or those that showed signs of severe illness and pain (hunched back, lack of grooming) were euthanized.

### Measuring fungal burden in mice

Fungal burden was determined in consistent lung, spleen, liver, and kidney tissue portions that were harvested and weighed at various time points post-infection. Tissue sections included: three right lung lobes, namely the superior, middle and lower lobes; a three-quarter section from the posterior extremity of the spleen; the entire liver except for the left lobe (used for histology); and the right kidney. After euthanasia, these organ sections were harvested aseptically and stored in 1–2 ml 0.1% PBS-Tween80 on ice followed by homogenization. Ten-fold serial dilutions of the homogenates were made in 0.1% PBS-Tween80 and 100 μl was plated in duplicate at various dilutions on BHI agar. Plates were incubated at 37˚C for 10–12 days for colony counting. Colony-forming units (CFU) per gram of tissue were calculated and presented to represent fungal burden and dissemination. The remaining lung and spleen homogenates, after plating, were centrifuged at 10 000 RPM for 10 min at 4˚C and the supernatants were collected, stored in aliquots and frozen at -80˚C for further cytokine measurement by ELISA.

### Tissue histopathology assessment

Consistent tissue sections of the lung (left lobe) were collected and immediately fixed in 4% (v/v) formaldehyde in PBS to preserve the tissue before embedding in paraffin wax and cut into 2–5 μm sections. Staining was done using hematoxylin and eosin (H&E) to view the alveolar spaces and inflammatory cells and Grocott's Methenamine Silver stain (GMS) (Sigma-Aldrich, USA) to stain fungal cells in infected tissue. Photomicrographs of stained tissue sections were captured using a Nikon Eclipse 90i microscope and DS-Ri2 high-performance camera (Nikon) and Nikon NIS-Elements imaging software (v4.30.01 DU1).

### Lung cellular recruitment by flow cytometry

Lung single-cell suspensions, 1 x $10^6$ cells/reaction, were stained with myeloid cell markers of neutrophils, inflammatory monocytes, eosinophils, alveolar macrophages, dendritic cells, and natural killer cells and blocked with 1% rat serum and 1% anti-FcγRII/III. Cells were identified as follows: neutrophils (CD11b$^+$, LY6G$^+$, LY6B$^+$), inflammatory monocytes (CD11b$^+$, LY6B$^+$, SiglecF$^{-neg}$), eosinophils (CD11b$^+$, SiglecF$^+$, LY6B$^{-neg}$), alveolar macrophages (CD11c$^+$, SiglecF$^+$, F4/80$^+$) and CD11b$^+$ dendritic cells (LY6G$^{-neg}$, CD11c$^+$, CD11b$^+$). Separate staining for natural killer (NK) cell populations in both mouse strains; gating for NK cells: (CD3$^{-neg}$, NK1.1$^+$). Stained, and fixed cells were acquired using the LSR Fortessa (BD Immunocytometry Systems, San Jose, CA, USA) and BD FACS Diva software (v6.0). Flowjo software (v10.0.7) (Treestar, Ashland, OR USA) was used for post-acquisition analysis and cell population determination. Refer to S1 Fig for gating strategies and S1 Table for a detailed list of antibodies used.

### Cytokine analysis from lung homogenates

The supernatant from lung and spleen homogenates was used to measure the presence of various cytokines that were released in infected tissue. The following BD OptEIA mouse ELISA kits were used according to the manufacturer's instructions: IFN-γ, IL-1β, TNF, IL-6, IL-10, IL-12p40 and IL-12p70. IL-17 (purified anti-mouse IL-17A clone: TC11-18H10.1; biotin anti-mouse IL-17A clone: TC11-8H4) and GM-CSF (purified anti-mouse GM-CSF; biotin anti-mouse GM-CSF) ELISA antibodies and standards were purchased separately from Biolegend.

### Serum total antibody ELISAs

Total IgM, IgG1, IgG2a, IgG2b, IgG3 and IgA antibody levels from both naïve and infected mice were measured in sera by coating Nunc-Immuno Maxisorb 96-well plates with unlabeled goat anti-mouse capture antibody at 1:500 dilution. Following blocking and incubation with sera samples, total antibody isotype titers were detected with alkaline phosphatase-conjugated rat anti-mouse antibodies at 1:1000 dilution. All antibodies used were produced by Southern Biotech.

### Statistical analysis

Two-tailed student's t-tests or One-way analysis of variance (ANOVA) tests were used to determine statistical significance unless otherwise indicated. Results are graphically represented as mean ± SD or SEM, with statistical significance represented as *: $p = 0.05$; **: $p = 0.01$; ***: $p = 0.001$ and ****: $p = 0.0001$. All experiments were independently repeated at least once unless otherwise indicated.

## Results

### Wild-type mice develop *Emergomyces africanus* infection

To establish a mouse model of disseminated disease for studying host-pathogen interactions, we first determined whether yeast cells of *E. africanus* could infect different mouse strains. A liquid fungal suspension was prepared from yeast cells of *E. africanus* isolated from a patient with advanced HIV disease. Susceptibility to infection was investigated in wild-type mice from three different genetic backgrounds: BALB/c, C57BL/6 and Sv129. As previously described for other fungi, all mice received a non-invasive intratracheal inoculation of between 0.5 to 1x$10^5$ yeast cells/50 µl [12]. Survival curves showed that all wild-type mice were susceptible to this high-dose infection (Fig 1A). Progressive weight loss occurred in all wild-type mice from 6-days post-infection (Fig 1B). Yeast infection caused pulmonary as well as disseminated infection in all wild-type mouse strains by two-weeks post infection, shown by organisms cultured

at 37˚C on brain heart infusion agar (BHI) plates in lung, spleen, liver, and kidney. Representative plates are shown in Fig 1C and 1D.

## Disseminated emergomycosis disease parameters in C57BL/6 wild-type mice

As many gene-deficient mice have been created on a C57BL/6 background, we focussed most of our subsequent studies on this strain, starting with a time-kinetic study of progression of

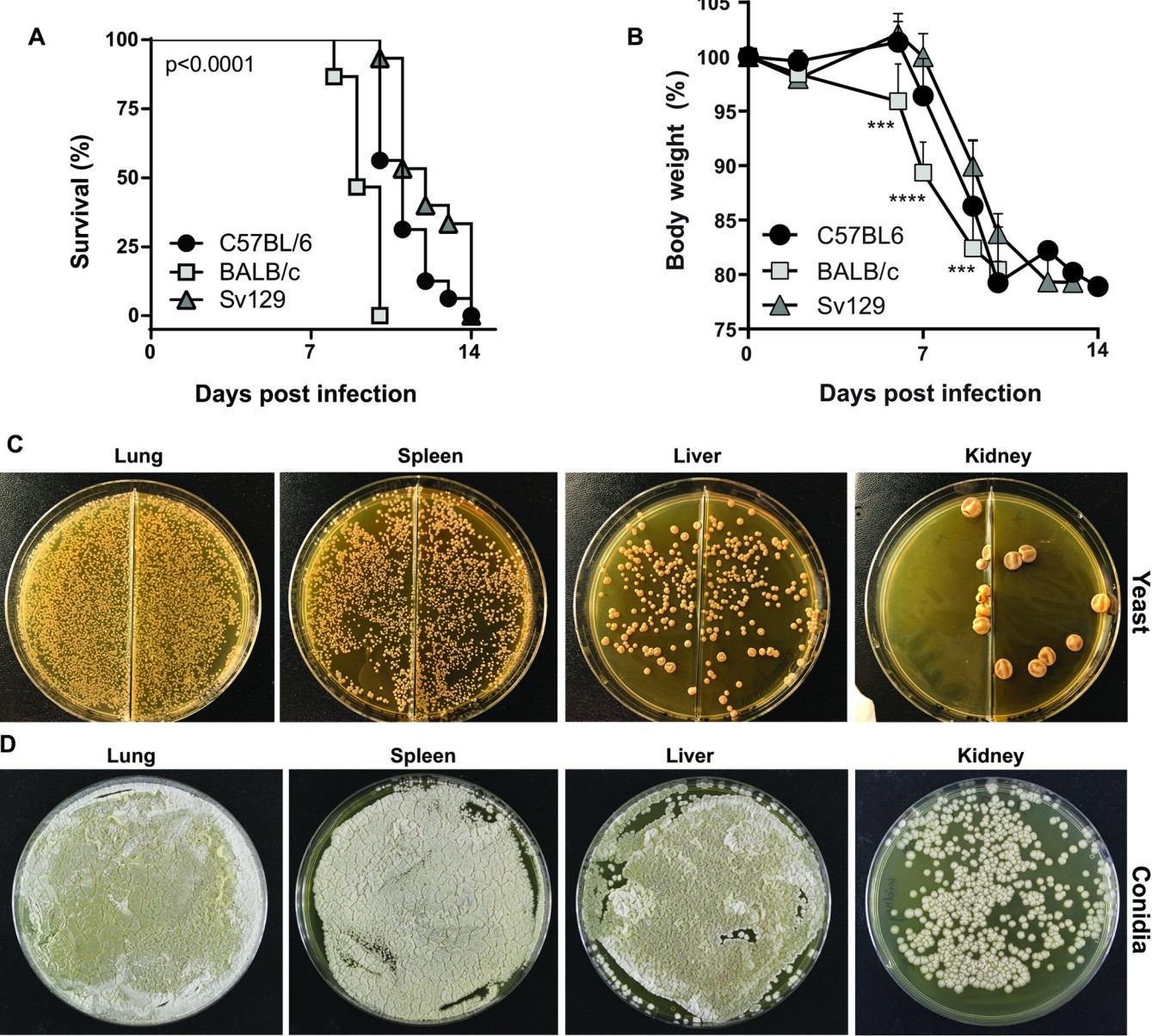

**Fig 1. *Emergomyces africanus* yeast cause disseminated infection in wild-type mice.** Three wild-type mouse strains C57BL/6, BALB/c and Sv129 were infected with 0.5-1x10$^5$ yeast cells in 50 μl PBS via oropharyngeal instillation. (A) A high dose infection resulted in 100% mortality (determined when mice reached 20% weight loss) by 14 days post infection although BALB/c mice were significantly more susceptible to infection when compared to C57BL/6 and Sv129 mice; p<0.0001. (B) Weight loss of mouse strains measured. Data represents two independent pooled experiments, n = 8 /group per experiment, with mean ± SD, ***p<0.001 and ****p<0.0001, (one-way AVOVA). (C) Disseminated fungal infection was confirmed in the lung, spleen, liver, and kidney by two weeks post-infection and yeast cells grew by plating homogenized organs on brain heart infusion (BHI) agar and incubated at 37˚C for two-three weeks. (D) Mold phase organisms were also isolated from tissue by culturing SDA plates at 29˚C for 2–3 weeks. WT = wild-type.

disease. Due to the higher dose resulting in mortality (described above), we infected mice with a lower dose (i.e., 1x10$^3$ *E. africanus* yeast cells/50 µl) via non-invasive intratracheal inoculation and euthanised the mice at weekly intervals. Uninfected, naïve mice were used as a comparison to determine *E. africanus*-specific responses. At the lower dose of 1x10$^3$ yeast cells/ 50 µl, C57BL/6 wild-type mice showed transient weight loss at 14- to 17-days post-infection when compared to uninfected mice (Fig 2A). Fungal organisms could be cultured from the lung, spleen, liver, and kidney (quantified as CFU per gram of respective tissue, Fig 2B). The fungal burden was detected in the lungs within 7 days post-infection and disseminated to other organs by two weeks post-infection. Except for the spleen, the fungal burden diminished and was undetectable by Day 28 in all organs, suggesting the mice could control infection in these tissues. Infected lung lobes were enlarged with macroscopic evidence of lesion formation, particularly at two- and three-weeks post-infection (Fig 2C). Haematoxylin and eosin (H&E) staining of the left lung lobes showed patchy or localized areas of infection and cellular involvement, resembling granuloma-like formations evident from two-weeks post-infection, waning by four weeks (Fig 2D). High magnification H&E and Grocott's methenamine silver stain (GMS)-stained sections for yeast localisation are shown in S1 and S2 Figs. Conidia were also able to cause disseminated disease with detectable organisms in lung, spleen, liver and kidney at day 14 post-infection (S3 Fig).

## T-helper type 1-mediated immune response to *Emergomyces africanus*

To further characterise the immune response to infection, we characterised cytokine levels in the lung and spleen during the kinetic study. Cytokine levels were compared with the levels obtained for uninfected mice (UI). Of interest, IFN-γ, IL-1β and IL-6 lung cytokine levels were significantly increased in the lung at both two- and three-weeks post-infection. These levels peaked by Day 21 post-infection and then returned to levels comparable to uninfected animals (see Fig 3 for the statistical description of cytokine production). Lung IL-12p40 was significantly increased at Day 14, Day 21 and Day 28 post-infection while IL-17 production was significantly higher at Day 14 post-infection (Fig 3). There was no notable change in lung TNF, IL-12p70 and IL-10 levels at any time point during infection (Fig 3).

During the kinetic study, blood was collected from mice by cardiac puncture immediately after euthanasia. Total blood serum antibody production over time was evaluated by antibody ELISA and compared to serum antibody levels in uninfected mice. Serum IgM, IgG1 and IgG2a antibody production was significantly increased at Day 21 and Day 28 post-infection when compared to uninfected mice (Fig 4).

## Increased susceptibility of immunocompromised mice to *E. africanus*

Since the majority of *E. africanus* cases are diagnosed in patients with advanced HIV disease (AIDS) and thus very low CD4$^+$ T-lymphocyte counts, a Rag-1-deficient (devoid of mature T and B lymphocytes) murine model of infection was established to study the infection in an immunocompromised host. Rag-1$^{-/-}$ mice were infected via non-invasive intratracheal inoculation with 1x10$^3$ yeast cells/ 50 µl and survival was compared to C57BL/6 wild-type mice. All Rag-1$^{-/-}$ mice succumbed to infection by Day 20 post-infection (Fig 5A) and the increased mortality was associated with a significant increase in fungal burden at Day 14 post-infection in all organs tested (Fig 5B). Histopathology demonstrated that diffuse cellular infiltrates were observed in infected lung tissue of C57BL/6 and Rag-1$^{-/-}$ mice at Day 7 post-infection (Fig 5C). At Day 14, both C57BL/6 and Rag-1$^{-/-}$ mice developed similar pathology with granuloma-like structures. Increased magnification of lung tissue areas confirmed a concentration of yeast cells (positive GMS staining) within the granuloma-like structures when compared to

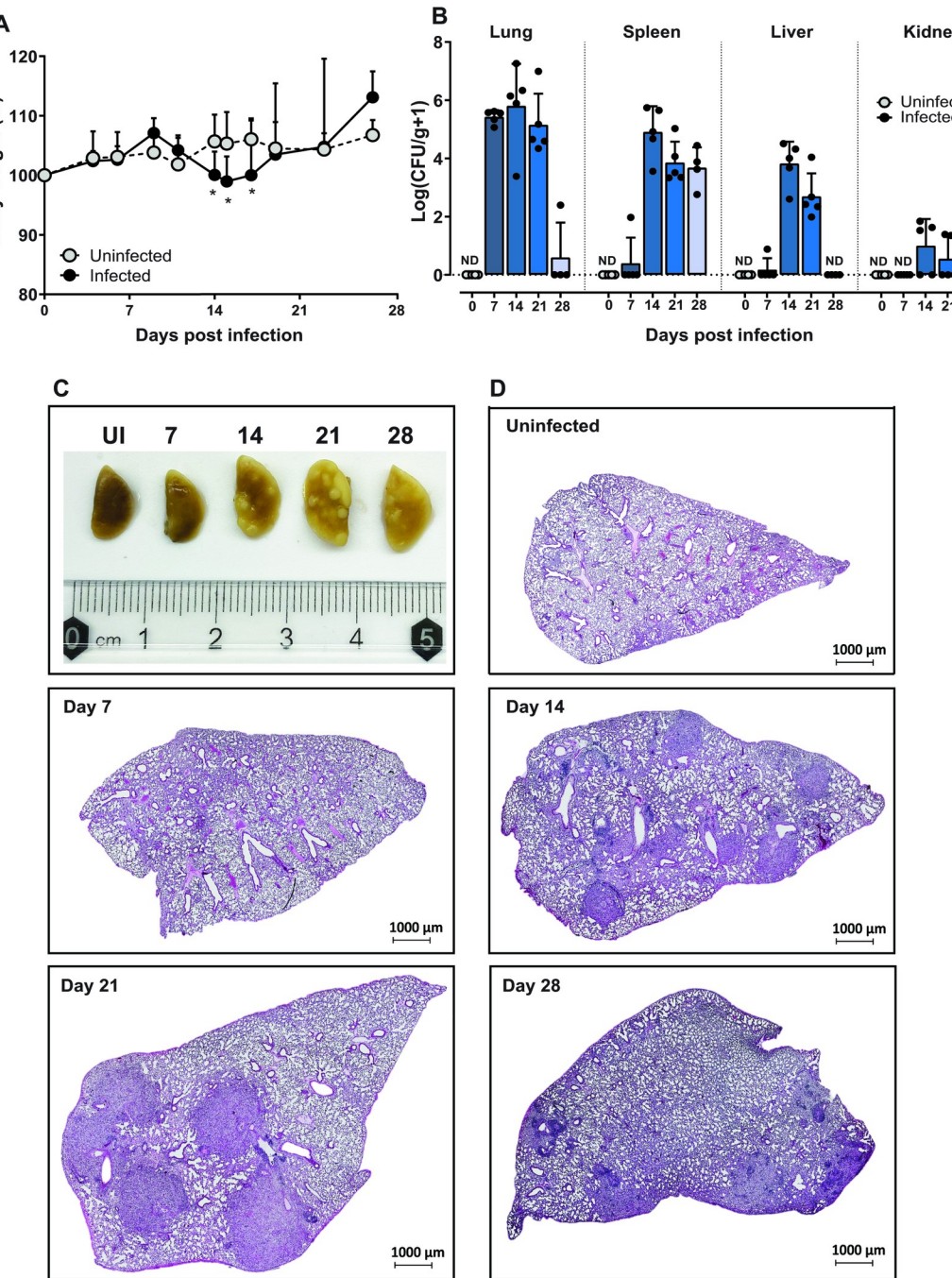

**Fig 2. Emergomycosis disease profile in wild-type mice.** C57BL/6 mice were infected with $1 \times 10^{3}$ yeast cells in 50 μl PBS and euthanized at one-week intervals post-infection for four weeks. Uninfected/naïve (UI) mice were used as a negative control. (A) C57BL/6 mice showed a transient weight loss compared to uninfected mice. (B) Infection resulted in the dissemination of organisms from the lung to the spleen, liver, and kidney shown by CFU per gram tissue. (C) Macroscopic image of fixed lobes showed lesion/granuloma formation, at Day 14 and Day 21 post-infection. (D) Representative photomicrographs of lung sections stained with H&E with original magnification: 2X. Data represents two individual experiments, n = 5 mice/group per experiment. UI: uninfected, H&E: hematoxylin and eosin.

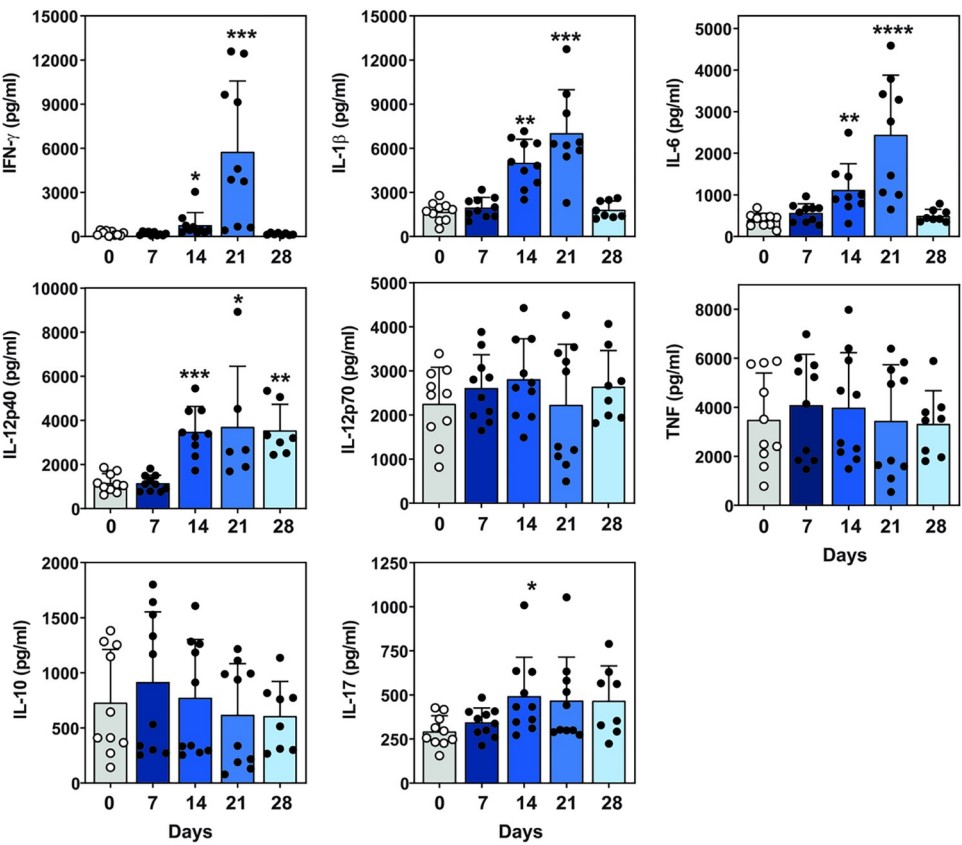

**Fig 3. Cytokine profile of infected lung tissue over time.** C57BL/6 mice were infected with 1x10³ yeast cells in 50 μl PBS and euthanized at one-week intervals post-infection. Supernatant from homogenized lung was used to test for cytokine levels using ELISA for, IFN-γ, IL-1β, IL-6, IL-12p40 and IL-17. Data represents two independent pooled experiments, n = 4–5 mice/group per experiment, with mean ± SD, *p<0.05, ***p<0.001 and ****p<0.0001 (one-way ANOVA using Dunnett's multiple comparisons test), C57BL/6 vs. uninfected mice. UI: uninfected.

surrounding tissue (S4 Fig). These data demonstrate the location of yeast cells, while the CFU data quantifies fungal burden.

## Susceptibility of Rag-1-deficient mice associated with reduced cellular recruitment and IFN-γ production

Since Rag-1⁻/⁻ mice had significantly increased fungal burden at two-weeks post-infection, we determined if there were differences in cellular recruitment to infected lung tissue in these mice. A myeloid cell flow cytometry panel was established and optimised for the infection model using C57BL/6 and Rag-1⁻/⁻ mice infected via non-invasive intratracheal inoculation with 1x10³ yeast cells/ 50 μl. Infected lungs were harvested at Day 14 post-infection and single cell suspension prepared for cell staining (S5, S6 Figs and S1 Table). Representative scatter plots are shown for each identified cell population in both wild-type and Rag-1⁻/⁻ mice (Fig 6A). Despite exacerbated disease and increased CFUs, Rag-1⁻/⁻ mice showed a significant increase in the frequency (%) of alveolar macrophages, neutrophils, and NK cells (Figs 6B and S7). Interestingly there was a significant decrease in eosinophils. Both inflammatory mono-cytes and dendritic cells were not different from wild-type mice. Notably, when measuring cytokine production by ELISA assay from lung tissue, IFN-γ was the only cytokine affected in Rag-1⁻/⁻ mice compared to C57BL/6 mice (Fig 7).

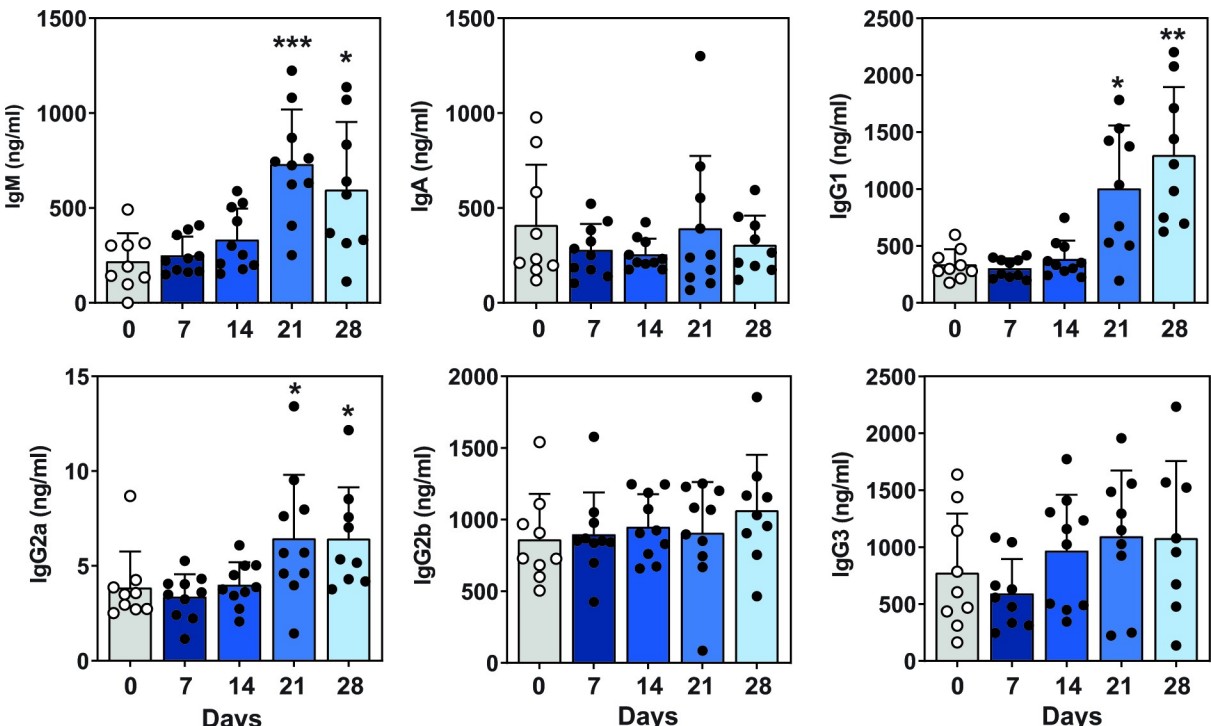

**Fig 4. Serum total antibody production over time.** Blood was collected during the disease kinetic studies from Day 7—Day 28 post-infection. Blood serum antibody levels were determined by ELISA and compared to antibody levels from uninfected mice. Data represents two independent pooled experiments, n = 4–5 mice/group per experiment, with mean ± SD and *p<0.05, **p<0.01 and ***p<0.001 (one-way ANOVA using the nonparametric Kruskal-Wallis test and Dunn's multiple comparisons test), C57BL/6 vs. uninfected mice. UI: uninfected.

## Discussion

This study reports the first pulmonary *Emergomyces* infection model in mice. We showed that following pulmonary infection, *E. africanus* disseminated to the spleen, liver, and kidneys within two weeks, and was cleared from the lung, liver, and kidney (but not spleen) by four weeks. Infection triggered a T helper-type 1 immune response supported by the detection of pro-inflammatory cytokines such as IFN-γ, IL-1β, IL-6, IL-12p40 and IL-17 in the supernatant of lung homogenates. Next, we used immunocompromised (Rag-1-deficient) mice, to identify immunologic cascades essential for controlling infection. This work advances our understanding of the nature and function of the immune system in emergomycosis and sets the foundation for future development of diagnostics and therapies for the disease.

Our data shows for the first time the susceptibility of experimental mice to the yeast (and conidia) form of a clinically isolated *E. africanus* strain. We provide a description of the disease kinetics and immune responses leading to fungal clearance in the lung and how these are compromised in Rag-1-deficient mice. To date *E. africanus* infections appear to be limited to immunocompromised patients with advanced HIV disease; however, with improved serological and/or antigen testing and therefore surveillance, detection is likely to increase in other groups of people.

Unlike other dimorphic fungal pathogens, an animal reservoir for *Emergomyces* spp. is yet to be found, despite efforts in screening 26 different species of small mammals [13]. Our data has shown that infection with the yeast (and conidia) form is possible in three distinct mouse strains: C57BL/6, BALB/c and Sv129. During survival studies in which mice were euthanised once reaching a humane endpoint of 20% weight loss, all mice eventually succumbed to

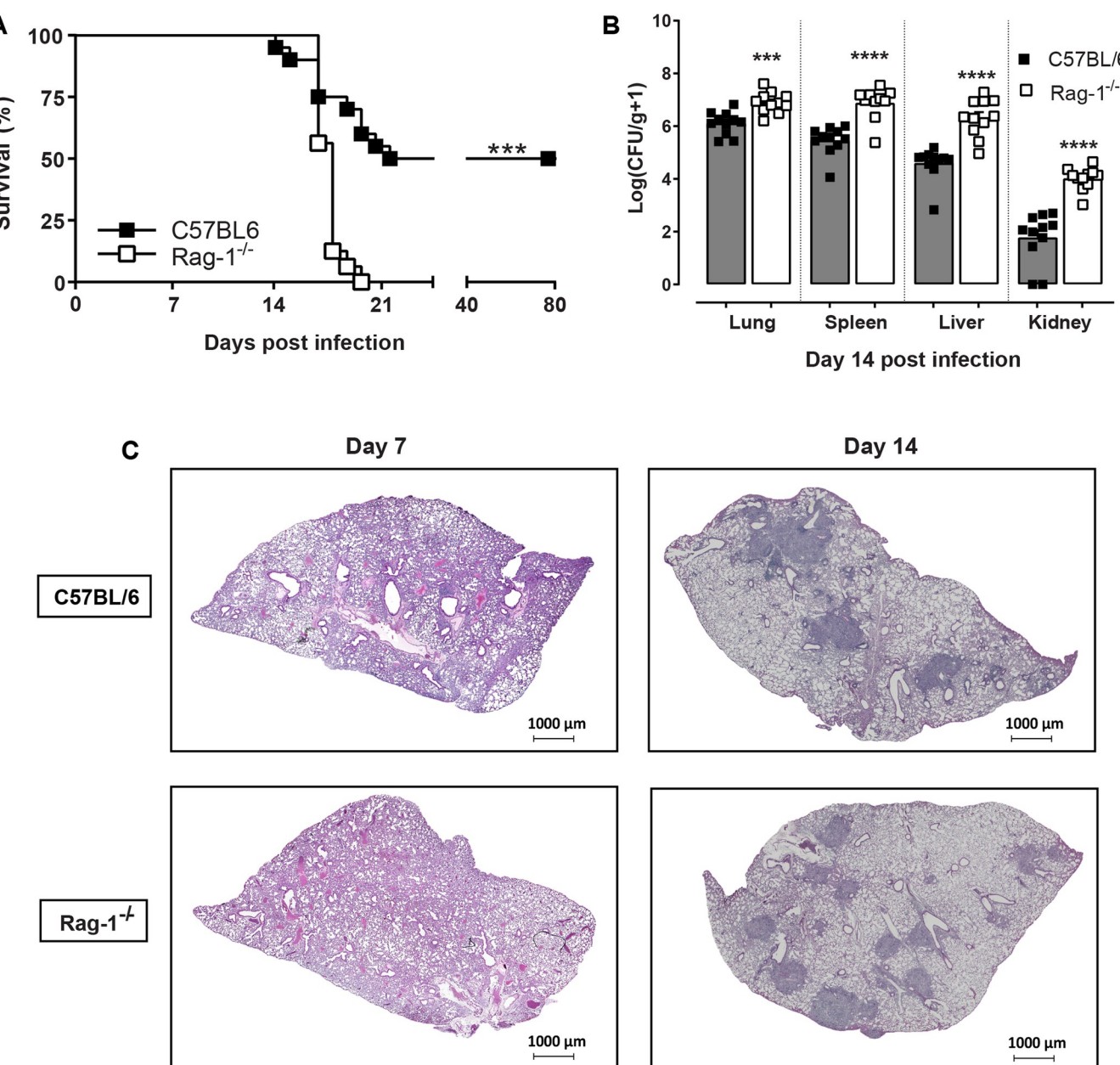

**Fig 5. Immunocompromised Rag-1-deficient mice are susceptible to *E. africanus* infection.** C57BL/6 and Rag-1[-/-] mice were infected with 1x10³ yeast cells and (A) Survival was compared. (B) In addition, infected C57BL/6 and Rag-1[-/-] mice were killed at Day 14 post infection and fungal burden was determined by CFU/gram of tissue in lung, spleen, liver and kidney. (C) Representative photomicrographs are shown of lung sections stained with H&E from C57BL/6 and Rag-1[-/-] mice at Day 14 post infection with original magnification: 2X. Data represents two independent pooled experiments, with n = 8–10 mice/group for survival and 5–6 mice /group for CFU and histology, mean ± SD, ***p<0.001 and ****p<0.0001 C57BL/6 vs Rag-1[-/-] (student t-test or one-way AVOVA).

infection at the higher does. For lower dose experiments, wild-type mice that did not need to be euthanized were able to clear the infection. Mice of different genetic backgrounds have been reported to respond differently to certain infection models. This is genetically determined as mice carry varied immunoregulatory genes within alleles of the H2 region [14]. These genes regulate the efficiency of antigen processing, cytokine production, as well as the expression and functioning of cytokine receptors. Therefore, C57BL/6 mice are prone to Th1- responses

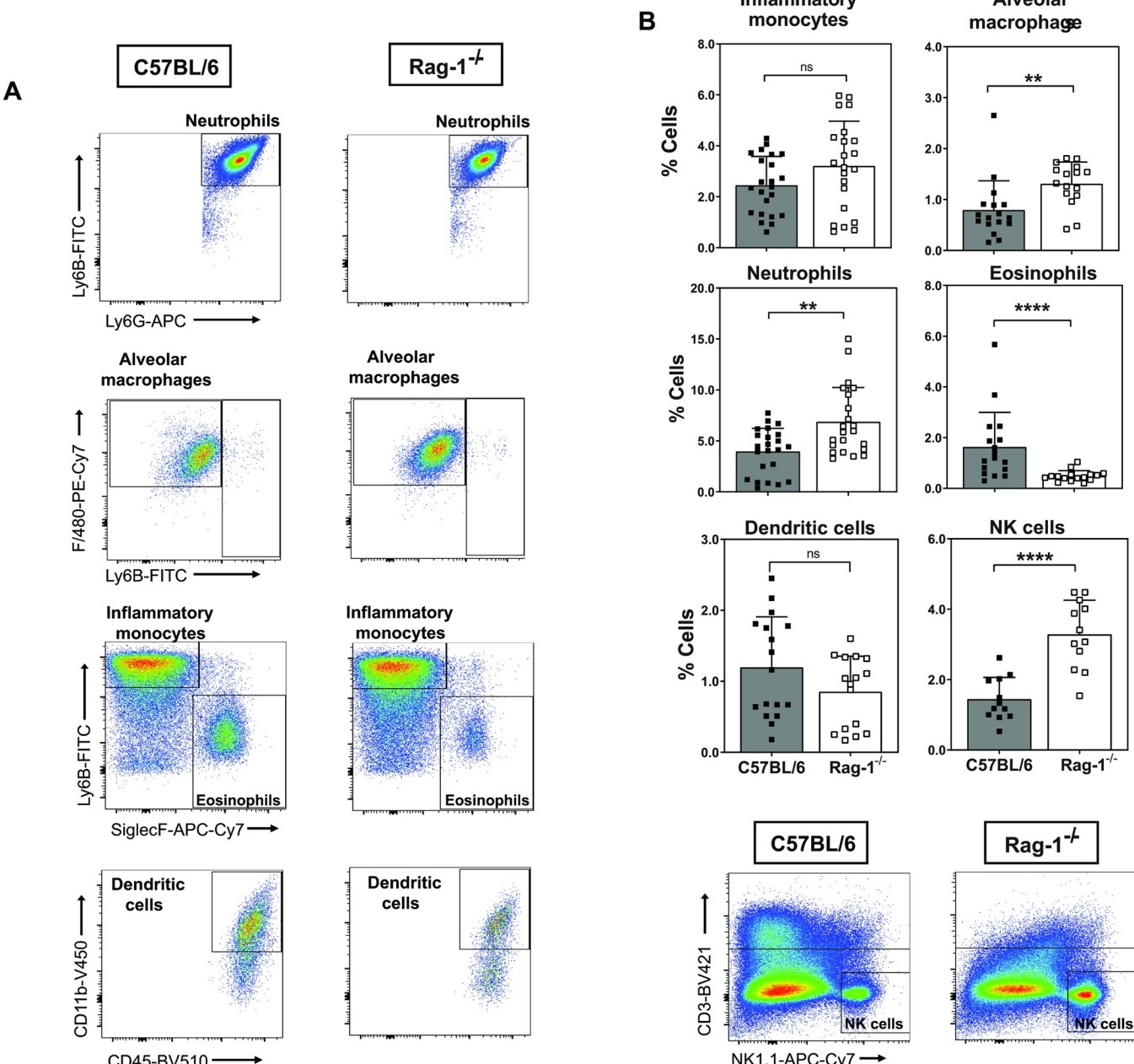

**Fig 6. Myeloid cell recruitment to infected lung tissue after *E. africanus* infection.** C57BL/6 and Rag-1$^{-/-}$ mice were infected with 1x10$^3$ yeast cells and lungs harvested at Day 14 post infection. (A) Representative myeloid cell population scatter plots from both C57BL/6 and Rag-1$^{-/-}$ mice. (B) Percentage cells shown for myeloid cell populations identified as follows: neutrophils (CD11b$^+$, LY6G$^+$, LY6B$^+$), inflammatory monocytes (CD11b$^+$, LY6B$^+$, SiglecF$^{neg}$), eosinophils (CD11b$^+$, SiglecF$^+$, LY6B$^{-neg}$), alveolar macrophages (CD11c$^+$, Siglecf $^+$, F4/80$^+$) and CD11b$^+$ dendritic cells (LY6G$^{-neg}$, CD11c$^+$, CD11b$^+$). Separate staining for NK cell populations in both mouse strains; (CD3$^{-neg}$, NK1.1$^+$). Data represents two pooled experiments, n = 4–5 mice/group per experiment, mean ± SD, **p<0.01 and ****p<0.0001 C57BL/6 vs. Rag-1$^{-/-}$ (student t-test). FMO: fluorescence minus one; NK: natural killer; ns = not significant.

and BALB/c mice Th2-prone [14]. Considering that the mice of interest for future studies are on a C57BL/6 background, and that *Emergomyces* appears to be an intracellular pathogen (likely to require Th1 responses for clearance) next we determined the disease progression and parameters in C57BL/6 mice. Mice infected with a lower dose over four weeks showed significant weight loss at 14 to 17-days post-infection when compared to uninfected mice but mice

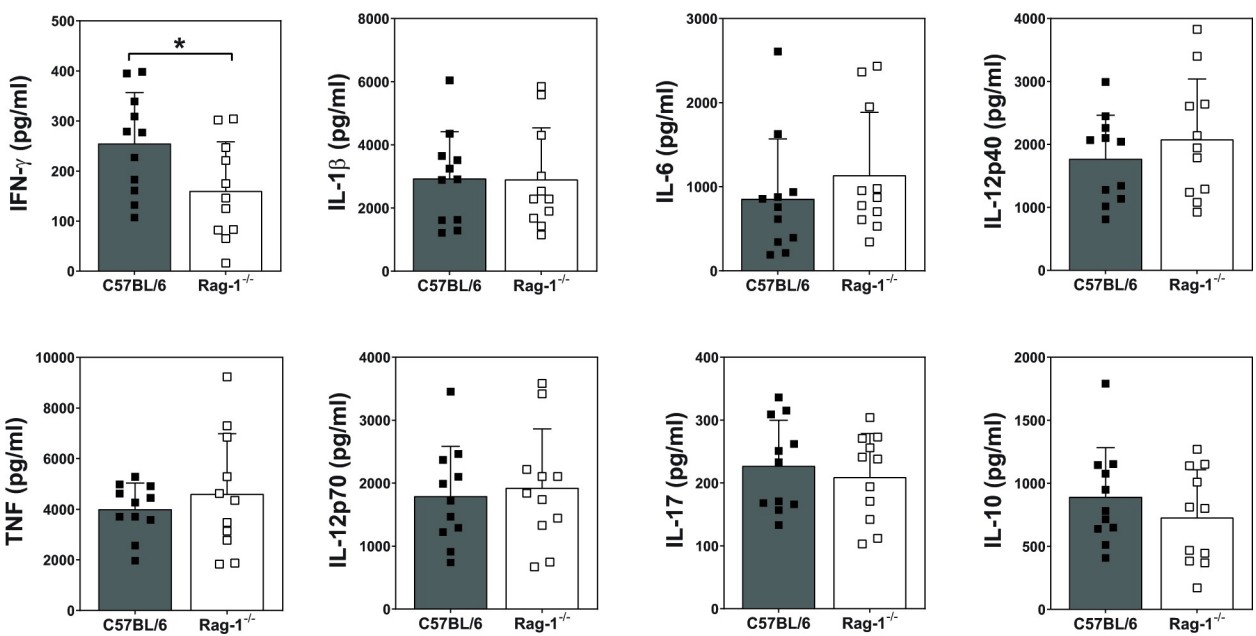

**Fig 7. Lung cytokine production in Rag-1-deficient mice in response to *E. africanus* infection.** Homogenized lung supernatant was used to test for cytokine levels at Day 14 post infection using ELISA. Data represents two pooled individual experiments, mean ± SD, n = 5–6 mice/group per experiment, and *p<0.05 C57BL/6 vs Rag-1$^{-/-}$, student t-test.

that survived were able to clear infection. Fungal propagules disseminated to multiple organs from the lung infection and the fungal burden seemed to peak at two weeks after which it was cleared from most organs.

Organ lesions and granulomas are described as a major interface between the host and fungal pathogens. H&E staining of lung sections during our kinetic study showed granuloma-like formations detected from two-weeks post-infection. Based on observations in other thermally dimorphic pathogens we speculate that granuloma formation during emergomycosis is induced by the infection of macrophages by yeast cells within the lung tissue, causing an infiltration of immune cells [15]. Granuloma formation is a beneficial response, required to localize fungal replication, limit dissemination, and protect the lung from extensive tissue damage. The reduced lung pathology in these studies coincided with a decrease in lung fungal burden by four-weeks post-infection. Interestingly, we saw an increase in cell debris noted in lung histology sections at three- and four weeks post-infection which may be indicative of cellular apoptosis and killing of yeast cells with possible necrosis occurring at the resolving granuloma sites. By the end of week four, the granuloma-like structures had resolved, which may have been facilitated by macrophages clearing apoptotic and yeast-infected cells.

Studies of the immune response to other thermally-dimorphic fungi show that innate immune cells, such as macrophages and dendritic cells, cooperate with the adaptive immune cells, such as CD4$^+$ and CD8$^+$ T cells, to control infection. The studies showed that early in infection, when alveolar macrophages interact with fungal cells, essential pro-inflammatory cytokines were produced: IL-12, TNF, IL-1, IFN-γ and GM-CSF. These cytokines were required for the activation of protective adaptive immune responses [16–18] Similarly, our study showed a significant increase in lung IFN-γ, IL-1β, IL-6, IL-12p40 and IL-17 cytokine production. However, these cytokines were only increased later in infection and with no difference in the lung at one-week post-infection, despite a large fungal burden. This data suggests that *E. africanus* may not elicit a pronounced innate immune response, unlike other

opportunistic fungal pathogens. Our study showed a steady significant increase in lung IL-12p40 from two- to four-weeks post-infection. In other dimorphic infection models, IL-12 production has been shown to be produced by macrophages and dendritic cells, importantly linking both innate and adaptive immunity. IL-12 production has been shown to be essential in adaptive immune responses to dimorphic fungi, leading to the promotion and generation of Th1 cells and cytokines (such as IFN-γ production by T cells) and subsequent protection against disseminated infection. Interestingly, our data showed no significant increase in TNF levels in lung supernatant during the *E. africanus* kinetic study. In contrast, TNF has been shown to play an important role in controlling *H. capsulatum* infection. Increased production of TNF was detected from granuloma infiltrating cells, primarily from macrophages [15]. In addition, studies using anti-TNF treatment were shown to reactivate latent *H. capsulatum* in some patients, highlighting an important role for TNF in mediating immune responses, controlling yeast infection and dissemination [19,20]. Perhaps the importance of TNF in response to *E. africanus* occurs earlier and would have been missed in our kinetic studies starting at one-week post-infection. Therefore, future studies could include cytokine detection in lung supernatant or bronchoalveolar lavage fluid within the first few days following infection to determine acute inflammatory immune responses to *E. africanus* infection. Our data demonstrated an increase in IL-17 production at two-weeks post-infection. However, there is conflicting data on the role of IL-17 in fungal clearance. On the one hand, increased IL-17 production by CD4⁺ T cells and activated macrophages has been shown to induce macrophage effector function during *H. capsulatum* infection [15]. In contrast, C-type lectin receptor-deficient mice with a decrease in lung-specific IL-17 were still able to clear infection. Therefore, we hypothesize that although IL-17 was increased, it may not be essential and rather that Th1 responses are central in driving clearance.

Antibodies have shown disparate biological effects in fungal infections, but the production of IFN-γ and opsonizing antibodies promote Th1 cells that activate phagocytes at sites of infection. Therefore, we measured serum antibody levels in our time-course study. IgM, IgG1 and IgG2a were increased at three- and four-weeks post-infection. Interestingly mice that were susceptible to *Paracoccidioides* infection have been correlated with preferential secretion of IgG2b-specific antibodies [21] rather than IgG2a and IgG1 which were shown to be correlated with improved disease outcomes. This may suggest a protective role for significantly increased IgG1 and IgG2a against *E. africanus*. Based on these findings it may be prudent to determine if antibodies are playing an essential role in controlling *E. africanus* infection by performing further studies in B cell-deficient mice.

Given the susceptibility of immunocompromised patients with advanced HIV disease to *E. africanus*, our next objective was to establish the mouse model using Rag-1-deficient mice. Rag-1⁻/⁻ mice lack mature T- and B-cells due to defective V(D)J recombination thus halting T- and B-cell differentiation at an early stage. We showed Rag-1⁻/⁻ mice to be more susceptible to *E. africanus* compared to C57BL/6 wild-type mice. This was associated with significantly increased fungal burden at two-weeks post-infection, as well as significantly higher dissemination to other organs including the spleen, liver, and kidney. Rag-1⁻/⁻ mice showed successful "granuloma" formation in the lungs at two-weeks post-infection. This could be attributed to the fact that macrophages have been shown to be the dominant cell type in fungal-mediated granuloma formation [15]. Interestingly, the only significant difference in lung cytokine production was a decrease in IFN-γ. IFN-γ has been shown to play a major role in dimorphic fungal immunity, with IFN-γ being crucial for the activation of macrophages, which play a central role in controlling infection. For example, IFN-γ-deficient mice were shown to be highly susceptible to *H. capsulatum* infection. [22,23]. The source of IFN-γ production is likely from CD4⁺ T cells that drive macrophage activation and fungal clearance. Our work demonstrates

that infected Rag1$^{-/-}$ mice were able to produce granuloma-like formations in the lungs. However, due to the lack of infiltrating CD4$^+$ and CD8$^+$ T cells, macrophages were probably unable to effectively kill intracellular yeasts which led to enhanced disseminated infection and death. While IFN-γ production was significantly decreased in Rag-1$^{-/-}$ mice, considering the enhanced susceptibility, one would anticipate a more exaggerate decrease in this cytokine. An explanation is that the increase in lymphoid NK cells produce IFN-γ. However, this is insufficient to arm all macrophages and inhibit the intracellular growth of yeasts. This together with the absence of CD4$^+$ and CD8$^+$ infiltrates surrounding granuloma formation may be the cause of increased dissemination and susceptibility of Rag-1$^{-/-}$ mice. IFN-γ immune therapy may be a promising area of research in the context of immunocompromised hosts.

We showed the myeloid cellular infiltrates in infected lung tissue at Day 14 post-infection of both wild-type and Rag-1$^{-/-}$ mice. Despite a significant increase in the percentage of neutrophils, alveolar macrophages and NK cells, Rag-1$^{-/-}$ mice developed exacerbated disease with increased fungal burden in all organs tested. Perhaps this increased cellular infiltration contributed to pathology and disease progression. While there was no significant difference in dendritic cells and inflammatory monocytes, the percentage of eosinophils was significantly reduced. Although neutrophils have been shown to be vital in the defense against some fungal infections such as *Candida* spp. and *Aspergillus* spp., dimorphic fungi such as *H. capsulatum*, *Blastomyces dermatitidis* and *Coccidioides immitis* do not rely on neutrophils [24]. Interestingly, dendritic cells have been shown to reduce the transformation of *H. capsulatum* from conidia to yeast form, therefore limiting the dissemination [25]. Dendritic cells have also been shown to kill *H. capsulatum* yeast cells by phagosome-lysosome fusion causing degradation of internalized yeasts [25]. Dendritic cells are also known to be an important link between innate and adaptive immunity as they can recognize and phagocytose fungi as well as present antigens to T cells. Dendritic cells may play an important role in wild-type mice by recognizing and phagocytosing *E. africanus* yeasts, resulting in IL-12 production and antigen presentation to T cells. Since there was only a subtle decrease in dendritic cells recruited to Rag-1$^{-/-}$ mouse lungs, with no decrease in lung IL-12 production; we hypothesize that IL-12 is produced by macrophages during *E. africanus* infection in Rag-1$^{-/-}$ mice. While we can use these observations to understand the host response during infection and how a weakened immune response may promote disease, we can only speculate the outcome during advanced HIV disease. Perhaps this could provide an explanation as to why we did not observe cutaneous lesions in mice as seen in human infection. This may be specific to advanced HIV patients with emergomycosis. Future work could include humanized mice infected with HIV or establishing a dermal infection model. Despite these limitations, the animal model of *E. africanus* is an indispensable step forward in understanding the pathogenesis and immune components required for clearing infection.

## Conclusion

Despite the prevalence and high mortality rates of invasive dimorphic fungal infections, they continue to be misdiagnosed or underdiagnosed and are largely understudied. This work has laid the foundation for a deeper understanding of *Emergomyces* spp. infection and provided valuable insight into disease kinetics and immune components involved in fungal clearance mechanisms in experimental mice. Case study reports indicate that people living with advanced HIV disease are more likely to develop serious symptomatic and disseminated emergomycosis requiring hospitalization. Late diagnosis leads to high fatality rates despite the initiation of amphotericin B treatment, with surviving patients requiring lengthy maintenance antifungal treatment (1–2 years). Here, we report that *E. africanus* infection drives a Th1

immune response in C57BL/6 mice. Importantly we showed that reduced IFN-γ production was associated with increased disease severity in immunocompromised Rag-1-deficient mice. Our model could be further utilized to explore new therapeutic approaches. Research focusing on immune therapy and possible vaccine development for endemic dimorphic fungal pathogens will have a tremendous impact on the treatment and management of the disease.

## Supporting information

**S1 Fig. Histopathology of disease progression in wild-type mice.** C57BL/6 mice were infected with $1x10^3$ yeast cells in 50 μl PBS and euthanized at weekly intervals post-infection. Sections were stained with H&E for inflammation and represented photomicrographs are shown for 10x and 20x magnification. (B) Representative images of lung sections stained with GMS at 10x and 20x magnification.
(PDF)

**S2 Fig. Fungal distribution in the lungs of wild-type mice.** C57BL/6 mice were infected with $1x10^3$ yeast cells in 50 μl PBS and euthanized at weekly intervals post-infection. Sections were stained with GMS for fungal organisms and representative photomicrographs are shown for 10x and 20x magnification. GMS: Grocott methenamine silver.
(PDF)

**S3 Fig.** *Emergomyces africanus* **conidia cause disseminated infection in wild-type mice.** C57BL/6 mice were infected with $1x10^3$ conidia cells in 50 μl PBS and euthanized at Day 14 post-infection. Conidia infection resulted in the dissemination of organisms from the lung to the spleen, liver, and kidney shown by CFU per gram tissue. Data represents two independent pooled experiments, n = 6 mice per experiment.
(PDF)

**S4 Fig. Histopathology and fungal distribution in the lungs of Rag-1-deficient mice.** C57BL/6 and Rag-1$^{-/-}$ mice were infected with $1x10^3$ yeast cells and euthanized at Day 14 post infection. Representative images of lung sections from C57BL/6 and Rag-1$^{-/-}$ mice stained with H&E and GMS at 10x and 20x magnification. H&E: hematoxylin and eosin; GMS: Grocott methenamine silver.
(PDF)

**S5 Fig. Myeloid flow cytometry gating strategy for infected lung tissue.** C57BL/6 and Rag-1$^{-/-}$ mice were infected with $1x10^3$ yeast cells. Lungs were harvested at Day 14 post-infection. The myeloid gating strategy for infected lung tissue was initiated by excluding cell debris and gating for single cells. Live leukocytes were identified as CD45$^+$ and Zombie Red-. Myeloid cell populations were gated as follows: neutrophils (CD11b$^+$, LY6G$^+$, LY6B$^+$), inflammatory monocytes (CD11b$^+$, LY6B$^+$, SiglecF$^{-neg}$), eosinophils (CD11b$^+$, SiglecF$^+$, LY6B$^{-neg}$), alveolar macrophages (CD11c$^+$, SiglecF$^+$, F4/80$^+$) and CD11b$^+$ dendritic cells (LY6G$^{-neg}$, CD11c$^+$, CD11b$^+$) (B) (see below).
(PDF)

**S6 Fig. Natural killer cell flow cytometry gating strategy.** C57BL/6 and Rag-1$^{-/-}$ mice were infected with $1x10^3$ yeast cells. Lungs were harvested at Day 14 post-infection. Separate staining was done to identify NK cell populations in both mouse strains; gating for NK cells: (CD3$^{-neg}$, NK1.1$^+$). FMO: fluorescence minus one; NK: natural killer.
(PDF)

**S7 Fig. Absolute number of myeloid cell recruitment to infected lung tissue after *E. africanus* infection.** C57BL/6 and Rag-1$^{-/-}$ mice were infected with 1x10$^3$ yeast cells and lungs harvested at Day 14 post-infection. Absolute cell numbers shown for myeloid cell populations identified as follows: neutrophils (CD11b$^+$, LY6G$^+$, LY6B$^+$), inflammatory monocytes (CD11b$^+$, LY6B$^+$, SiglecF$^{-neg}$), eosinophils (CD11b$^+$, SiglecF$^+$, LY6B$^{-neg}$), alveolar macrophages (CD11c$^+$, Siglecf $^+$, F4/80$^+$) and CD11b$^+$ dendritic cells (LY6G$^{-neg}$, CD11c$^+$, CD11b$^+$). Separate staining for NK cell populations in both mouse strains; gating for NK cells: (CD3$^{-neg}$, NK1.1$^+$). Data represents three-four pooled experiments, n = 4–5 mice/group per experiment, mean ± SD, $^*$p<0.05 and $^{***}$p<0.001 C57BL/6 vs. Rag-1$^{-/-}$ (student t-test). FMO: fluorescence minus one, NK: natural killer.
(PDF)

**S1 Table. Flow cytometry antibody information.** This table provides a list of FACS antibodies used to identify cell types.
(PDF)

## Acknowledgments

We thank Dr Barbra Topless for her initial training in *Emergomyces* spp. culture techniques. We are grateful to Faried Abbass and Rodney Lucas for their valuable technical assistance and the UCT animal unit staff for the running and maintenance of the facility. We would like to thank Lizette Fick for her excellent technical assistance in histology services and Prof Dhiren Govender for providing a description of histopathology.

## Author Contributions

**Conceptualization:** Ilan S. Schwartz, Nelesh P. Govender, Gordon D. Brown, Jennifer Claire Hoving.

**Investigation:** Maxine A. Höft, Lucian Duvenage, Sumayah Salie, Roanne Keeton.

**Project administration:** Sumayah Salie.

**Resources:** Alfred Botha, Nelesh P. Govender, Gordon D. Brown, Jennifer Claire Hoving.

**Supervision:** Gordon D. Brown, Jennifer Claire Hoving.

**Validation:** Gordon D. Brown, Jennifer Claire Hoving.

**Visualization:** Maxine A. Höft.

**Writing – original draft:** Maxine A. Höft.

**Writing – review & editing:** Maxine A. Höft, Lucian Duvenage, Sumayah Salie, Roanne Keeton, Alfred Botha, Ilan S. Schwartz, Nelesh P. Govender, Gordon D. Brown, Jennifer Claire Hoving.

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
