## [Decision Letter · Decision Letter 0]

25 Sep 2023

Dear Mrs. Hoving,

Thank you very much for submitting your manuscript "The pathogenesis of experimental Emergomycosis in mice." for consideration at PLOS Neglected Tropical Diseases. As with all papers reviewed by the journal, your manuscript was reviewed by members of the editorial board and by several independent reviewers. The reviewers appreciated the attention to an important topic. Based on the reviews, we are likely to accept this manuscript for publication, providing that you modify the manuscript according to the review recommendations. 

Sincerely,

Ahmed Hassan Fahal, FRCS, FRCSI, FRCSG, MS, MD, FRCP(London)

Academic Editor

Marcio Rodrigues

Section Editor

Reviewer's Responses to Questions

**Key Review Criteria Required for Acceptance?**

**Methods**

-Are the objectives of the study clearly articulated with a clear testable hypothesis stated?

-Is the study design appropriate to address the stated objectives?

-Is the population clearly described and appropriate for the hypothesis being tested?

-Is the sample size sufficient to ensure adequate power to address the hypothesis being tested?

-Were correct statistical analysis used to support conclusions?

-Are there concerns about ethical or regulatory requirements being met?

Reviewer #1: The study was well designed and the authors have managed to establish a novel murine model of E. africanus disease providing critical insights into the host immune components required for eliminating the infection. No concerns about the ethical or regulatory requirement met.

The methods were suitable to reach a sensible and measurable outcomes.

Reviewer #2: -Are the objectives of the study clearly articulated with a clear testable hypothesis stated? Yes

-Is the study design appropriate to address the stated objectives? Not entirely

-Is the population clearly described and appropriate for the hypothesis being tested? yes

-Is the sample size sufficient to ensure adequate power to address the hypothesis being tested? Yes

-Were correct statistical analysis used to support conclusions? Yes

-Are there concerns about ethical or regulatory requirements being met? No concerns

**Results**

-Does the analysis presented match the analysis plan?

-Are the results clearly and completely presented?

-Are the figures (Tables, Images) of sufficient quality for clarity?

Reviewer #1: The analysis of data displayed matches the analysis plan. The results are extensive but clear and one cytokine (IL 10) results needs to be displayed in the abstract because it is key cytokine antagonising interferon gamma. Good quality figures and images.

Reviewer #2: -Does the analysis presented match the analysis plan? Yes

-Are the results clearly and completely presented? Yes

-Are the figures (Tables, Images) of sufficient quality for clarity? Yes

**Conclusions**

-Are the conclusions supported by the data presented?

-Are the limitations of analysis clearly described?

-Do the authors discuss how these data can be helpful to advance our understanding of the topic under study?

-Is public health relevance addressed?

Reviewer #1: Extracted from the results.

Reviewer #2: -Are the conclusions supported by the data presented? Not entirely

-Are the limitations of analysis clearly described? Yes

-Do the authors discuss how these data can be helpful to advance our understanding of the topic under study? Yes

-Is public health relevance addressed? Yes

**Editorial and Data Presentation Modifications?**

Reviewer #1: IL 10 results can be presented in the abstract as key cytokine when addressing IFN-gamma and immunopathogenesis.

Reviewer #2: There is a spelling error on line 310.

**Summary and General Comments**

Reviewer #1: Well designed and conducted study. Acceptable for publication.

Reviewer #2: This is a study to establish a mouse model of pulmonary Emergomyces africanus infection that is observed in some HIV patients with advanced disease. To further understand the biology of E. Africanus infections, the authors utilize Rag 1 deficient mice. 

The overall strengths of this study were the novelty of the model, the use of different mouse models with Th1 and Th2- biased responses and some mechanistic insights from the Rag1 deficient mice. The data was clearly presented with impactful imagery. Indeed, the authors were able to establish infection in various tissues that persisted for 2 or more weeks, therefore this is a feasible experimental model. 

However, there were weaknesses in the study: the association with advanced HIV in the experimental model was not shown, therefore, I think that associations with HIV are only speculative as the authors did not infect the mice with HIV. There should be a consideration of the weaknesses of rodent models for HIV pathogenesis and how this impacts the presentation of other pathologies associated with HIV infection. Why did the authors only focus on immunoanalyses of innate immune cells? Especially given that they proceed to use Rag-1 deficient mice which have defective adaptive immune systems? What would be the importance of the significantly lower eosinophils in the Rag 1-/- mice that had worse disease? Also, it is noted that the cytokines are produced a week after infections despite a large fungal burden. How is this mediated and why is the response so delayed? Is there another player of the adaptive immune system that may be triggering these responses in the lung? How closely does the widespread infection in mice (in multiple tissues) compare to the human case? Was this a hyperbolic response? If so, how can one interpret results from these models in a human context? Was there a reason humanized mice (with a reconstitution of the human immune systems) were not used in these studies? What about the immune response in other tissues? Of interest is the persistent infection in the spleen, an immunoregulatory site... why was this evident? Some speculation on this would be of interest. A lot of comparisons are made with H. capsulatum infections in the discussion. These comparisons confuse the reader and sometimes I am not sure if the paper was to establish a murine model of E. africanus infections or to compare and contrast its pathology to known H. capsulatum infections. Why would we care about H. capsulatum infections anyway in the context of advanced HIV? Most patients with advanced HIV are also on ARVs. What would be the impact of ARV treatment on E. africanus infections?

PLOS authors have the option to publish the peer review history of their article (what does this mean?). If published, this will include your full peer review and any attached files.

Reviewer #1: No

Reviewer #2: No

Figure Files:

Data Requirements:

Reproducibility:

References

---

## [Decision Letter · Decision Letter 1]

7 Dec 2023

Dear Mrs. Hoving,

We are pleased to inform you that your manuscript 'The pathogenesis of experimental Emergomycosis in mice.' has been provisionally accepted for publication in PLOS Neglected Tropical Diseases.

Best regards,

Marcio L Rodrigues

Section Editor

---

## [Editor Report · Acceptance letter]

5 Jan 2024

Dear Mrs. Hoving,

We are delighted to inform you that your manuscript, "The pathogenesis of experimental Emergomycosis in mice.," has been formally accepted for publication in PLOS Neglected Tropical Diseases.

Best regards,

Shaden Kamhawi

co-Editor-in-Chief

Paul Brindley

co-Editor-in-Chief
